# Design and Synthesis of New Sulfonic Acid Functionalized Ionic Liquids as Catalysts for Esterification of Fatty Acids with Bioethanol

**DOI:** 10.3390/molecules28135231

**Published:** 2023-07-05

**Authors:** Thu Huong Nguyen Thi, Jiřina Koutecká, Pavel Kaule, Luboš Vrtoch, Václav Šícha, Jan Čermák

**Affiliations:** Faculty of Science, Jan Evangelista Purkyně University in Ústí nad Labem, Pasteurova 15, 400 96 Ústí nad Labem, Czech Republic; thu-huong.nguyen-thi@ujep.cz (T.H.N.T.); pavel.kaule@ujep.cz (P.K.); lubos.vrtoch@ujep.cz (L.V.); vaclav.sicha@ujep.cz (V.Š.)

**Keywords:** ionic liquids, esterification catalysis, sulfonic acid, fatty acids, bioethanol

## Abstract

In this study, three types of sulfonic acid group functionalized ionic liquids (SAILs) with a different number of catalytic groups and lipophilicity were synthesized and characterized by FT-IR, NMR, and MS analyses. Their catalytic activities were studied in a model esterification of oleic acid with ethanol; heating in a microwave reactor was also used. The experimental results indicated that SAIL, with the lipophilic alkyl chain, performed the best due to its increased solubility in the reaction mixture. Microwave heating was found to be more effective than conventional heating. Recycling experiments show that these novel SAILs can be reused without significant loss of the catalytic activity.

## 1. Introduction

The availability of crude oil is expected to decrease in future despite new oil fields being discovered. As a reaction to this serious issue, renewable biofuels, including biodiesel [1], are being developed. Biodiesel is usually a mixture of fatty acid esters with the most common being methyl (FAME) and ethyl (FAEE) esters. Fatty acids can come from various sources but the current trend is a feedstock from waste cooking, frying or nonedible oil, and also animal fat (mutton, chicken, ostrich etc.). Such feedstock, however, contains free fatty acids (FFA) which must be subjected to esterification before base-catalyzed transesterification to FAME or FAEE is carried out because of unwanted simultaneous saponification of FFA. Therefore, the esterification of FFA must be studied to find new approaches including efficient new catalysts.

Traditionally, the industry relies on heterogeneous catalysts [1,2,3,4,5,6] but there are other possibilities. Ionic liquids (IL) were extensively studied over several decades [7,8] and have attracted the attention of biodiesel producers [9] as catalysts among other applications (solvents, extraction media). Comparably simple imidazolium hydrogen sulfates were studied [10,11] but recently, more sophisticated catalysts were published such as IL supported on metal oxides, polymers, carbon, metal-organic frameworks, magnetic composites [12,13,14], or polyamidoamine dendrimers [15]. Kinetic and thermodynamic studies [16] and the application of diethyl carbonate as a transesterification reagent [17] were published in processes with ionic liquids as the catalysts. Recently, ultrasound intensification of IL-catalyzed algal biodiesel production was published [18] with multiobjective optimization.

The IL used are most frequently ammonium or imidazolium salts in which the acidic site is represented by a sulfonic acid group. Xian and coworkers synthesized various 3-sulfopropylammonium (including imidazolium) methanesulfonates and tosylates and showed them to catalyze the conversion of several FFA into their esters with several alcohols [19]. Similar catalysts were used by Xia et al. [20] in a study of the acidity–activity relationship with the conclusion that the presence of the SO_3_H- group in the catalysts is essential. Miao, Lv et al. used 3-sulfopropylimidazolium hygrogensulfates for the transesterification of rapeseed oil [21]. Further research focused on increasing the number of SO_3_H- groups [22,23,24]. For example, Ma et al. [25] prepared disulfonic-functionalized acid ionic liquids as efficient catalysts for biodiesel synthesis and carefully examined the role of hydrogen bonds. Recently Qiu et al. [26] synthesized so called self-solidification ionic liquids starting from low-cost amines of the oligo(ethylenediamine) type. Multi SO_3_H- functionalization allowed solidification of IL after the reaction because of two types of SO_3_H- groups: the bonding ones forming zwitterions with amine groups and the free ones responsible for catalytic activity. Lin, Qiu et al. [27] later carried out esterification of oleic acid with methanol with IL of this type and a reached high conversion of 97.6%. The IL catalysts were characterized by a number of analytical techniques and, for optimization, the Box–Behnken surface methodology was used. In a continuation of this interesting approach, Lin, Qiu et al. [28] synthesized self-solidifying quarternary phosphonium-containing ionic liquids which showed good reusability in methanol esterification of oleic and several other fatty acids.

Many ionic liquids are rather polar compounds due to their ionic character. However, making, the usually cationic part, more lipophilic could be beneficial for esterification reaction. With this idea in mind, Qin et al. [29] synthesized long-chain Brønsted acid ionic liquids for biodiesel synthesis from FFA and short-chain alcohols with very good reusability of up to nine runs. The ionic liquids were expected to function similarly to phase transfer catalysts. Qiu et al. [30] later named IL with lipophilic chains amphiphilic and published an extensive study of the dependence of biodiesel yield on the length of the chain. Chains C3-C10, C12, C14, C16, and C16 were examined with the maximum yield being at C12. The concept of higher lipophilicity was also used by Lozano et al. [31] who used ionic liquid with a C16 aliphatic chain for the extraction of biodiesel after the biocatalytic transformation of algal oil.

The last concept that we would like to mention in connection with biodiesel production is microwave (MW) heating. ILs have excellent microwave absorbance because of their ionic character; short-chain alcohols are also expected to strongly absorb MW irradiation. Therefore, their mixtures are able to transfer energy rapidly under microwave fields which were used, e.g., in direct transesterification of algal biomass to biodiesel [32,33]. Studies have shown the benefits of microwave heating, such as a shorter reaction time, and efficient and volumetric heating [34,35,36,37]. Bölük and Sönmez selected a combination of IL catalysts and used the microwave method as a process to accelerate the esterification of oleic acid which can be used in biodiesel production. The reaction reached equilibrium in 30 min, the methyl oleate yield was 85.7% and the conversion 94.1% [38]. Yan et al. [39] produced biodiesel by acidic IL catalyzed esterification of oleic acid with methanol under a combination of MW and ultrasonic irradiation. The combined irradiation can intensify the process and significantly shorten the reaction time to 15 min with the oleic acid conversion being 97.85%.

The catalytic stability and simple separation of the used catalyst from the reaction medium are the most important factors for the reusability of the catalyst, hence the most important factors in designing a new catalyst. In this work, we synthesized several sulfonic acid group functionalized ionic liquids (SAIL) with a different number of sulfonic acid catalytic groups and varying lipophilicity as potential catalysts for the esterification of fatty acids recycled from waste oil with bioethanol. These catalysts would combine the advantages of a homogeneous reaction (high activity, selectivity) with the operational advantages of heterogeneous systems (easy separation, easy recyclability, reusability) and very stable activity (Figure 1).

## 2. Results and Discussion

### 2.1. Synthesis of SAILs

Figure 1 shows the routes for the synthesis of the SAIL catalysts **1**–**8**. 2-Pentadecyl-1*H*-benzimidazole and 2,2′-(butan-1,4-diyl)bis(1*H*-benzimidazole) were obtained through condensation of o-phenylenediamine with palmitic or adipic acid resp. Then 1,3-propanesultone was reacted through bimolecular nucleophilic substitution (S_N_2 opening) with imidazole or benzimidazole activated by NaH to yield the IL precursors. Imidazole gave the only product sodium [1-(3-sulfopropyl)-1*H*-imidazol-3-ium-3-yl]propane-1-sulfonate, but each of benzimidazoles gave two sodium sulfonate products, which then were separated by column chromatography. Finally, a reaction with H_2_SO_4_ yielded the desired SAIL catalysts.

In the reaction of 1,3-propanesultone with imidazole and benzimidazole activated by NaH, first, molecule 1,3-propanesultone bonded to atom N in the heterocycle very readily in an hour. Bonding of the second molecule of 1,3-propanesultone to the heterocycle was more difficult. In the case of 2-pentadecyl-1*H*-benzimidazole, a small amount of product **4** was still present in the reaction mixture after 2 weeks of heating, and in the case of 2,2′-(butan-1,4-diyl)bis(1*H*-benzimidazole), only a small amount of product **7** was obtained.

### 2.2. Catalyst Evaluation for Oleic Acid Esterification with Ethanol

Oleic, and palmitic acids (75/25, *w*/*w*) esterification with ethanol was used as a model reaction for evaluating the catalytic activities of different SAILs. A fatty acids mixture was used as a mimic of waste oil with highly renewable bioethanol. The esterification conditions were chosen as follows: molar ratio ethanol (1 mL, 16.5 mmol)/fatty acids (0.32 mL, 1.0 mmol) according to the literature [11].

In the microwave esterification runs, catalyst loading varied from 0.1 to 0.3 equivalent to fatty acids, (the concentration range of SAIL was 0.1; 0.2; 0.3 mol/L). Reactions were monitored in 10 min intervals by TLC analysis, and conversions of oleic acid after 30 min microwave heating were determined by HPLC MS. In microwave esterification experiments, good catalytic activity of **SAIL 2** and **SAIL 5** was found at a concentration 0.1 mol/L, **SAIL 8** showed very small catalytic activity under those conditions. The catalytic activity of **SAIL 2** is best at a concentration 0.3 mol/L, the catalytic activity of **SAIL 5** is also observed best at this concentration but a concentration of 0.2 mol/L gives conversion almost as good as the former one. Reaction equilibrium was established after 20–30 min. Table 1 shows conversions of oleic acid after 30 min of microwave heating, determined by HPLC MS.

Conventional heating esterification experiments were carried out with 0.3 eq catalysts **SAIL 2**, **SAIL 5**, and **SAIL 8** and reactions were monitored in 2 h intervals by TLC analysis. The reaction temperature was set at 100 °C in the bath, in accordance with previous microwave experiments. TLC indicated reaction equilibrium after approximately 4 h. Conversions of fatty acids after 8 h heating were determined by HPLC MS as described in Table 2, respectively.

During the esterification reaction, **SAIL 2** and **SAIL 5** exhibited good miscibility with the reactants forming a homogeneous system during heating, and after cooling to room temperature, the SAIL catalysts were separated from the reaction mixture at the bottom of the flask. **SAIL 8** was not dissolved in the reaction mixture and formed a suspension, and that could be the reason why it showed very low catalytic activity under microwave heating.

For evaluating the catalytic efficiency and catalyst reusability, a series of recycling experiments with the same molar ratio ethanol to fatty acids 16.5:1, 30% molar loading of **SAILs 2**, **5**, **8**, and conventional heating at bath temperature 100 °C in 2 h. Generally, the catalysts have no effect on the chemical equilibrium. In the catalyzed esterification, SAIL catalysts are not reacting in a subsequent step, they are not consumed as reactants as the reaction proceeds, and then they cannot change the equilibrium concentrations. So, to compare those SAIL catalysts in catalytic efficiency, we determined conversion in a certain time before reaction equilibrium was established. The results from TLC and HPLC MS indicated that they are not very different in their catalytic efficiency. 

Five cycles of esterification were carried out under the optimum reaction conditions. After each cycle, the crude products were easily separated, removed from the reaction mixture and analyzed by HPLC MS. SAILs in the residue were reused in subsequent runs. As shown in Table 2, the FA conversions after 2 h varied in the range of 67–99%, indicating good to excellent catalyst reusability. The SAIL catalysts were used at least 5 times, and the only in the case of **SAIL 8** was there almost no change in the catalytic performance. Overall, SAILs were readily recycled, and are promising catalysts for the esterification of FA to biodiesel.

A comparison of the effectivity of our catalysts with the literature is shown in Table 3; there is also a large review [9] where additional data can be found. It is apparent that although the reaction conditions vary considerably, the literature data about conversion are grouped around 95–96%. Our best result—almost quantitative conversion (99.7%)—is thus outstanding.

## 3. Materials and Methods

### 3.1. Materials and Reagents

All reagents used in this study were supplied by Sigma–Aldrich, Acros Organics, Penta, and Lachema. Imidazole **1b** (99%, Sigma-Aldrich, Prague, Czech Republic), 1,3-propanesultone **1a** (97%, Acros Organics, Geel, Belgium), o*-*phenylenediamine **3a** (99.5%, Sigma-Aldrich), a mixture of oleic and palmitic acids (75/25 *w*/*w*, Lach-Ner, Neratovice, Czech Republic), adipic acid (pure, Lachema, Brno, Czech Republic), sodium hydride 60% suspension (Sigma–Aldrich), *N,N-*dimethylformamide (99.8%, Sigma-Aldrich), and sulphuric acid 96% (Penta, Prague, Czech Republic) were used to synthesise the ILs. 

The model esterification to biodiesel was carried out using bioethanol (96%) and fatty acids. Standard ethyl-oleate was used for the identification of the product by TLC. A silica gel matrix TLC aluminium plate without a fluorescent indicator was supplied by Sigma–Aldrich. All other solvents and reagents were HPLC MS grade and analytical reagent grade.

Dimethylformamide (DMF) was distilled under argon, hexane was dried with sodium and distilled under argon. Solution of HCl in methanol (approximately 2.7 M) was obtained through bubbling HCl (formed from the reaction of 16 g NaCl and 16 mL conc. H_2_SO_4_) into 100 mL methanol. Synthetic procedures were carried out under an inert atmosphere until acidic workup.

### 3.2. Characterization Methods

The structures of the ionic liquids and their intermediates were analyzed by IR and NMR spectroscopy and mass spectrometry. The supporting spectroscopic data can be found in the Appendix A).

Infrared spectra were measured on a Nicolet 6700 FT-IR spectrometer from Thermo Scientific with the ATR technique using a single reflection horizontal ZnSe crystal. The measurement was carried out at a resolution of 4 cm^−1^ and the number of scans was 150. The spectra were recorded and processed by the OMNIC 7.3 program.

^1^H and ^13^C-NMR spectra were obtained on a JEOL 400 MHz JNM-ECZ400R/M1 nuclear magnetic resonance spectrometer. Chemical shifts are reported in ppm and spin-spin coupling constants are presented in Hz.

Mass spectrometry (MS) spectra were performed on a Thermo Scientific Inc. LCQ Fleet Ion Trap instrument using electrospray (ESI) ionization with helium (6.0 Messer, Czech Republic) as a collision gas. Samples dissolved in 5% (*v*/*v*) aqueous acetic acid mixed with methanol (1:1, *v*:*v*) (concentrations approximately 100 ng·mL^−1^) were introduced to the ion source from a Hamilton syringe using an infusion of 15 μL·min^−1^, source voltage 5.0 kV, tube lens voltage −110.0 V, capillary voltage −35.0 V, capillary temperature 275 °C and N_2_ as a nebulizing sheath gas 15 p.d.u. In all cases, positive ions corresponding to the molecular ion were observed for the highest peak in the isotopic distribution plot. The isotopic distribution, and the whole shape of the spectra of all detected peaks were in agreement with the calculated spectra. The measurement range was 150 to 2000 a.m.u.

Conversions of esterification reactions were calculated from TIC chromatograms of fatty acids and their ethylesters measured on the system HPLC MS based on a Thermo Finnigan Surveyor HPLC (Autosampler Plus, LC Pump Plus) in tandem with an LCQ Fleet Ion Trap MS detector equipped with APCI ionization probe. Merck Purosphere Star column RP-8, 250 × 2 mm, 5 µm, and acetonitrile-water in 50 min long linear gradient from 60/40 (*v*/*v*) to 90/10 (*v*/*v*) in the flow of 300 µL/min as a mobile phase were used for simultaneous determination of fatty acids, and their respective ethylesters.

### 3.3. Synthesis of the Catalysts

#### 3.3.1. Brønsted Acidic Imidazolium-Based IL

*3-[1-(3-Sulfopropyl)-1H-imidazol-3-ium-3-yl]propane-1-sulfonate* **1**

A suspension of sodium hydride (60%, 1.52 g, 38 mmol) was weighed into a two-necked 250 mL flask fitted with a condenser, the apparatus was evacuated and filled with argon and the process was repeated twice. Sodium hydride suspension was washed 3 times with dried hexane (3 × 5 mL), then predistilled DMF (75 mL) and imidazole **1b** (2.50 g, 37 mmol) were added and the reaction mixture was stirred vigorously to form a clear solution of sodium imidazol-1-ide. The mixture was cooled in an ice bath for 15 min and 1,3-propanesultone **1a** (6.5 mL, 74 mmol) was added, then the reaction mixture was heated at 110 °C in an oil bath for 2 weeks. After that, approximately 60 mL of DMF was distilled off under reduced pressure of 130 °C, the residue was washed 3 times with diethyl ether (3 × 50 mL). The crude product was mixed with methanol (50 mL) and HCl solution in methanol was added until an acidic reaction. Flash chromatography on 20 g silica gel with elution by methanol gave 9.4 g (82%) of the white crystalline product; **mp** 214–219 °C; **^1^H-NMR** (D_2_O, 400 MHz) 1.18–2.25 (4H, m, NCH_2_C*H*_2_CH_2_), 2.83 (4H, t, *J* = 7.4 Hz, NCH_2_CH_2_C*H*_2_), 4.26 (4H, t, *J* = 7.2 Hz, NC*H*_2_CH_2_CH_2_), 7.46 (2H, s, -C*H*=C*H*-), 8.77 (1H, s, NC*H*=N); **^13^C-NMR** (D_2_O, 100 MHz) 25.09 (s, NCH_2_*C*H_2_CH_2_), 47.30 (s, NCH_2_CH_2_*C*H_2_), 47.93 (s, N*C*H_2_CH_2_CH_2_), 122.63 (s, -*C*H=*C*H-), 135.83 (s, N*C*H=N); **IR** (ATR, cm^−1^) 3154, 3100 (C-H, imidazole), 2950–2850 (C-H, (CH_2_)_3_), 1563, 1441 (C=C, C=N, imidazole), 1179, 1044 (S=O, -SO_3_H, -SO_3_^−^); ESI **MS** *m*/*z* [M + H]^+^ 313.14 (100), calculated for C_9_H_17_N_2_O_6_S_2_: 313.05.

*[1,3-di(3-Sulfopropyl)-1H-imidazol-3-ium] hydrogensulfate* **2**

Concentrated sulfuric acid (0.75 mL, 13 mmol) was added to [1-(3-sulfopropyl)-1*H*-imidazol-3-ium-3-yl]propane-1-sulfonate **1** (4.04 g, 13 mmol) in ethanol (50 mL), the mixture was stirred vigorously to form a clear solution, then heated in an oil bath at 80 °C for 1 h. After vacuum evaporation, repeated washing with diethyl ether and vacuum drying, 5.29 g (99%) of white solid product was obtained; **mp** 192–198 °C; **^1^H-NMR** (D_2_O, 400 MHz) 2.15–2.22 (4H, m, NCH_2_C*H*_2_CH_2_), 2.80 (4H, t, *J* = 7.4 Hz, NCH_2_CH_2_C*H*_2_), 4.23 (4H, t, *J* = 7.2 Hz, NC*H*_2_C*H*_2_CH_2_), 7.43 (2H, d, *J* = 2.0 Hz, -C*H*=C*H*-), 8.74 (1H, t, *J* = 2.0 Hz NC*H*=N); **^13^C-NMR** (D_2_O, 100 MHz) 25.05 (s, NCH_2_*C*H_2_CH_2_), 47.25 (s, NCH_2_CH_2_*C*H_2_), 47.89 (s, N*C*H_2_CH_2_CH_2_), 122.59 (s, -CH=CH-), 135.80 (s, NCH=N); **IR** (ATR, cm^−1^) 3150, 3096 (C-H, imidazole), 2950–2850 (C-H, (CH_2_)_3_), 1570, 1447 (C=C, C=N, imidazolium), 1220, 1168, 1023 (S=O, -SO_3_H, HSO_4_^−^); ESI **MS** *m*/*z* [M + H]^+^ 313.08 (65), calculated for C_9_H_17_N_2_O_6_S_2_: 313.05.

#### 3.3.2. Brønsted Acidic Benzimidazolium-Based IL

*2-Pentadecyl-1H-benzo[d]imidazole* **3b**

A mixture of palmitic acid (8.26 g, 32 mmol) and *o*-phenylenediamine **3a** (3.28 g, 32 mmol) was placed into a 250 mL round bottom flask with a magnetic stirrer. The apparatus was then evacuated and filled with argon and the process was repeated twice. The reaction mixture was heated at 185 °C and stirred vigorously for 30 min. The crude product was dissolved in 30 mL of ethanol with heating, a spoon of activated charcoal was added to the hot solution, the hot solution was filtered and a small amount of water was added to turbidity. The crystalization was carried out at room temperature. After suction filtration, repeated washing with a mixture of ethanol and water 1:1, and drying, 6.5 g of white crystalline product was obtained, yield 62%; **mp** 131–138 °C; **^1^H-NMR** (CDCl_3_, 400 MHz) 0.84 (3H, t, *J* = 6.8 Hz, C*H*_3_-), 1.14–1.28 (22H, m, CH_3_-C_11_*H*_22_C_3_H_6_-benzimidazole), 1.34 (2H, m, C_12_H_25_C*H*_2_C_2_H_4_-benzimidazole), 1.95 (2H, m, *J* = 7.7 Hz, C_13_H_27_C*H*_2_CH_2_-benzimidazole), 3.16 (2H, t, *J* = 7.6 Hz, C_13_H_27_CH_2_C*H*_2_-benzimidazole), 7.31 (2H, m, C*H*=C*H* (C5, C6 in benzimidazole)), 7.67 (2H, m, -C*H*= (C4, C7 in benzimidazole)), 11.90 (1H, bs, -N*H*-); **^13^C-NMR** (CDCl_3_, 100 MHz) 14.20 (s, *C*H_3_-), 22.76 (s), 27.89 (s), 28.24 (s), 29.32–29.78 (-*C*_n_H_2n_-), 32.00 (s, C_13_H_27_*C*H_2_CH_2_-benzimidazole), 53.51 (s, C_14_H_29_*C*H_2_-benzimidazole), 114.24 (s, -*C*H=, C4, C7 in benzimidazole), 124.29 (s, -*C*H=*C*H, C5, C6 in benzimidazole), 134.19 (s, >*C*= in benzimidazole), 154.72 (s, HN-*C*=N, C2 in benzimidazole); ESI **MS** *m*/*z* [M + H]^+^ 329.50 (100), calculated for C_22_H_37_N_2_: 329.30; **IR** (ATR, cm^−1^) 3430, 3300 (C-H, benzimidazole), 2918, 2848 (C-H, C_15_H_31_), 1625, 1541, 1457 (C=C, C=N, benzimidazole), 751, 722 (C-H, benzimidazole).

*3-(2-Pentadecyl-1-(3-sulfopropyl)-1H-benzo[d]imidazol-3-ium-3-yl)propane-1-sulfonate* **3**

SAIL **3** was preparated from 2-pentadecyl-1*H*-benzo[*d*]imidazole (3.30 g, 10 mmol), activated with sodium hydride (60% suspension, 0.52 g, 12.5 mmol), and 1,3-propanesultone **1a** (2.66 mL, 30 mmol) according to the procedure for the preparation of SAIL **1**. After chromatography on 70 g silica gel with elution by mixture of dichloromethane and methanol, 7:3 was obtained 4.2 g (73% yield) of oily product 3-(2-pentadecyl-1-(3-sulfopropyl)-1*H*-benzo[*d*]imidazol-3-ium-3-yl)propane-1-sulfonate **3**; **^1^H-NMR** (CD_4_OD, 400 MHz) 0.87 (3H, t, *J* = 6.6 Hz, C*H*_3_-), 1.19–1.44 (22H, m, CH_3_-C_11_*H*_22_C_3_H_6_-benzimidazole), 1.56 (2H, m, C_12_H_25_C*H*_2_C_2_H_4_-benzimidazole), 1.78 (2H, m, C_13_H_27_C*H*_2_CH_2_-benzimidazole), 2.35 (4H, m, NCH_2_C*H*_2_CH_2_), 2.97 (4H, t, *J* = 6.8 Hz, NCH_2_CH_2_C*H*_2_), 3.35 (2H, t, *J* = 8.2 Hz, C_14_H_29_C*H*_2_-benzimidazole), 4.71 (4H, t, *J* = 7.8 Hz, NC*H*_2_CH_2_CH_2_), 7.63 (2H, m, C*H*=C*H*, C5, C6 in benzimidazole), 8.00 (2H, m, -C*H*=, C4, C7 in benzimidazole); **^13^C-NMR** (CD_4_OD, 100 MHz) 13.10 (s, *C*H_3_-), 22.40 (s, NCH_2_*C*H_2_CH_2_), 23.49 (s, CH_3_-*C*H_2_-), 24.76 (s, CH_3_-CH_2_-*C*H_2_-), 27.61 (s, NCH_2_CH_2_*C*H_2_), 29.06–29.47 (m, -*C*_n_H_2n_-), 31.74 (s, C_14_H_29_*C*H_2_-benzimidazole), 44.02 (s, N*C*H_2_CH_2_CH_2_), 112.96 (s, -*C*H=, C4, C7 in benzimidazole), 126.51 (s, -*C*H=*C*H-, C5, C6 in benzimidazole), 131.34 (s, >*C*= in benzimidazole), 153.62 (s, HN-*C*=N, C2 in benzimidazole); ESI **MS** *m*/*z* [M + H]^+^ 573.56 (75), calcd for C_27_H_61_N_2_O_6_S_2_: 573.40; **IR** (ATR, cm^−1^): 3439, 3100 (C-H, benzimidazole), 2922, 2852 (C-H, C_15_H_31_), 1636, 1520, 1469 (C=C, C=N, benzimidazole), 1159, 1033 (S=O, -SO_3_H).

Together was isolated 1.3 g of 3-(2-pentadecyl-1*H*-benzo[*d*]imidazol-1-yl)propane-1-sulfonic acid **4**; **mp** 120–124 °C; **^1^H-NMR** (CDCl_3_, 400 MHz) 0.85 (3H, t, *J* = 6.8 Hz, C*H*_3_-), 1.10–1.42 (24H, m, CH_3_-C_12_*H*_24_C_2_H_4_-benzimidazole), 1.75–1.91 (2H, m, C_13_H_27_C*H*_2_CH_2_-benzimidazole), 2.32–2.52 (2H, q, NCH_2_C*H*_2_CH_2_), 2.98 (2H, t, *J* = 6.0 Hz, NCH_2_CH_2_C*H*_2_), 3.25 (2H, t, *J* = 7.8 Hz, C_14_H_29_C*H*_2_-benzimidazole), 4.72 (2H, t, *J* = 7.6 Hz, NC*H*_2_CH_2_CH_2_), 7.39–7.50 (2H, m, C*H*=C*H*, C5, C6 in benzimidazole), 7.70–7.78 (1H, m, -C*H*=, C4 in benzimidazole), 7.95–8.05 (1H, m, -C*H*=, C7 in benzimidazole); ESI **MS** *m*/*z* [M+H]^+^ 451.48 (100), calculated for C_25_H_43_N_2_O_3_S_1_: 451.30; **IR** (ATR, cm^−1^): 3300 (C-H, benzimidazole), 2915, 2849 (C-H, C_15_H_31_), 1736, 1665, 1568, 1518, 1470 (C=C, C=N, benzimidazole), 1170, 1030 (S=O, -SO_3_H).

*2-Pentadecyl-1,3-di(3-sulfopropyl)-1H-benzo[d]imidazol-3-ium hydrogen sulfate* **5**

SAIL **5** was preparated from 1-(3-sulfopropyl)-[(2-pentadecyl-1*H*-benzimidazol-3-ium)-3-yl]propane-1-sulfonate **3** (2.4 g, 4.2 mmol) and sulfuric acid (0.24 mL, 4.3 mmol) acording to the procedure for the preparation of SAIL **2**, to yield 2.8 g (100%) of oily product; **^1^H-NMR** (CD_4_OD, 400 MHz) 0.88 (3H, t, *J* = 6.6 Hz, C*H*_3_-), 1.23–1.50 (24H, m, CH_3_-C_12_*H*_24_C_2_H_4_-benzimidazole), 1.55 (2H, m, C_13_H_27_C*H*_2_CH_2_-benzimidazole), 1.80 (2H, m, C_13_H_27_C*H*_2_CH_2_-benzimidazole), 2.34 (4H, m, NCH_2_C*H*_2_CH_2_), 3.01 (4H, t, *J* = 6.8 Hz, NCH_2_CH_2_C*H*_2_), 3.34 (2H, t, *J* = 8.2 Hz, C_14_H_29_C*H*_2_-benzimidazole), 4.73 (4H, t, *J* = 8.0 Hz, NC*H*_2_CH_2_CH_2_), 7.64 (2H, m, C*H*=C*H*, C5, C6 in benzimidazole), 8.02 (2H, m, -C*H*=, C4, C7 in benzimidazole); **^13^C-NMR** (CD_4_OD, 100 MHz) 13.14 (s, *C*H_3_-), 22.39 (s, CH_3_-*C*H_2_-), 24.75 (s, CH_3_-CH_2_-*C*H_2_-), 24.80 (s, NCH_2_*C*H_2_CH_2_), 28.86 (s, -CH_2_CH_2_-benzimidazole), 29.03 (s, NCH_2_CH_2_*C*H_2_), 29.23–29.44 (-*C*_n_H_2n_-), 31.72 (s, C_13_H_27_*C*H_2_CH_2_-benzimidazole), 44.03 (s, N*C*H_2_CH_2_CH_2_), 112.96 (s, -*C*H=, C4, C7 in benzimidazole), 126.51 (s, *C*H=*C*H, C5, C6 in benzimidazole), 131.34 (s, >*C*= in benzimidazole), 153.59 (s, HN-*C*=N, C2 in benzimidazole); ESI **MS** *m*/*z* [M+H]^+^ 573.44 (100), calculated for C_28_H_49_N_2_O_6_S_2_: 573.30; **IR** (ATR, cm^−1^): 3439 (C-H, benzimidazolium), 2923, 2852 (C-H, C_15_H_31_), 1636, 1522, 1470 (C=C, C=N, benzimidazolium), 1168, 1023 (S=O, -SO_3_H, HSO_4_^−^).

*2,2′-(Butane-1,4-diyl)bis(1H-benzo[d]imidazole)* **6a**

2,2′-(butane-1,4-diyl)bis(1*H*-benzo[*d*]imidazole) was preparated from adipic acid (7.3 g, 0.05 mol) and *o*-phenylenediamine **3a** (10,8 g, 0.10 mol) acording to the procedure for the preparation of 2-pentadecyl-1*H*-benzo[*d*]imidazole. Recrystallization from ethanol-water yielded 3.7 g (26%) of white crystalline product; **mp** 258–265 °C; **^1^H-NMR** (DMSO-*d*_6_, 400 MHz) 1.81 (4H, m, benzimidazole-CH_2_-C*H*_2_-C*H*_2_-CH_2_-benzimidazole), 2.82 (4H, t, *J* = 6.4 Hz, benzimidazole-C*H*_2_-CH_2_-CH_2_-C*H*_2_-benzimidazole), 7.06 (4H, m, C*H*=C*H*, C5, C6 in benzimidazole), 7.41 (4H, m, -C*H*=, C4, C7 in benzimidazole), 12.16 (2H, bs, N*H* in benzimidazole); **^13^C-NMR** (DMSO-*d*_6_, 100 MHz) 27.70 (s, benzimidazole-CH_2_-*C*H_2_-*C*H_2_-CH_2_-benzimidazole), 28.77 (s, benzimidazole-*C*H_2_-CH_2_-CH_2_-*C*H_2_-benzimidazole), 118.78 (s, -*C*H=, C4, C7 in benzimidazole), 121.29 (s, *C*H=*C*H, C5, C6 in benzimidazole), 134.79 (s, >*C*= in benzimidazole), 155.43 (s, HN-*C*=N, C2 in benzimidazole); ESI **MS** *m*/*z* [M − H]^–^ 289.35 (100); [M+H]^+^ 291.50 (100),calculated for C18H19N4: 291.16; **IR** (ATR, cm^−1^): 3300–2500 (C-H, benzimidazole), 3051, 2932 (C-H, C_4_H_8_), 1621, 1529, 1453, 1415 (C=C, C=N, benzimidazole), 742 (C-H, benzimidazole).

*2,2′-(butan-1,4-diyl)bis[(1-(3-sulfopropyl)-1H-benzo[d]imidazol-3-ium-3-yl)propane-1-sulfonate]* **6**

and

*2-{4-[1-(3-sulfopropyl)-1H-benzo[d]imidazole-2-yl]butyl}-1-(3-sulfopropyl)-1H-benzo[d]imidazol-3-ium-3-ylpropanesulfonic acid* **7**

A preparation from 60% suspension of sodium hydride (1.0 g, 25 mmol), 1,4-bis(1*H*-benzo[*d*]imidazol-2-yl)butane (2.9 g, 10 mmol) and 1,3-propanesultone **1a** (4.4 mL, 6.1 g, 50 mmol) acording to the procedure for the preparation of SAIL **1** and following chromatography on 50 g silica gel, with elution by dichloromethane:methanol 4:1 then methanol, yielded 3.5 g (53%) of 2-{4-[1-(3-sulfopropyl)-1*H*-benzo[*d*]imidazole-2-yl]butyl}-1-(3-sulfopropyl)-1H-benzimidazol-3-ium-3-ylpropanesulfonic acid **7**; **mp** > 300 °C; **^1^H-NMR** (DMSO-*d*_6_, 400 MHz) 1.85; 2.12 (2H; 2H, m, benzimidazole-CH_2_-C*H*_2_-C*H*_2_-CH_2_-benzimidazole), 2.04 (6H, m, NCH_2_C*H*_2_CH_2_), 2.50 (6H, m, NCH_2_CH_2_C*H*_2_), 2.61 (4H, t, *J* = 6.6 Hz, -C*H*_2_-benzimidazole), 4.50 (2H, t, *J* = 7 Hz), 4.50 (2H, t, *J* = 7.8 Hz, NC*H*_2_CH_2_CH_2_), 4.67 (4H, t, *J* = 7.8 Hz, NC*H*_2_CH_2_CH_2_), 7.34 (2H, m, C*H*=C*H*, C5′, C6′ in benzimidazole), 7.58 (2H, m, C*H*=C*H*, C5, C6 in benzimidazole), 7.67; 7.79 (1H; 1H, m, -C*H*=, C4′, C7′ in benzimidazole), 8.05 (2H, m, C*H*=C*H*, C4, C7 in benzimidazole); ESI **MS** *m*/*z* [M+H]^+^ 657.12 (100), calculated for C_27_H_37_N_4_O_9_S_3_: 657.17; **IR** (ATR, cm^−1^) 3417 (C-H, benzimidazole), 2940 (C-H, C_4_H_8_), 1651, 1518, 1483 (C=C, C=N, benzimidazole), 1184, 1157, 1044 (S=O, -SO3H, -SO_3_^−^).

Together was isolated 0.2 g (3% yield) of 2,2′-(butan-1,4-diyl)bis[(1-(3-sulfopropyl)-1*H*-benzo[*d*]imidazol-3-ium-3-yl)propane-1-sulfonate] **6**; **mp** > 300 °C; **^1^H-NMR** (D_2_O, 400 MHz) 1.96 (4H, m, benzimidazole-CH_2_-C*H*_2_-C*H*_2_-CH_2_-benzimidazole), 2.23 (8H, m, NCH_2_C*H*_2_CH_2_), 2.95 (8H, t, *J* = 7.0 Hz, NCH_2_CH_2_C*H*_2_), 3.28 (4H, t, *J* = 6.4 Hz, -C*H*_2_-benzimidazole), 4.56 (8H, t, *J* = 8.2 Hz, NC*H*_2_CH_2_CH_2_), 7.53 (4H, m), 7.76 (4H, m); ESI **MS** *m*/*z* [M+H]^+^ 779.26 (100), calculated for C_30_H_43_N_4_O_12_S_4_: 779.14; **IR** (ATR, cm^−1^) 3423 (C-H, benzimidazole), 2931 (C-H, C_4_H_8_), 1728, 1641, 1517, 1464 (C=C, C=N, benzimidazole), 1161, 1039 (S=O, -SO_3_H, -SO_3_^−^).

*2-{4-[1-(3-sulfopropyl)-1H-benzo[d]imidazol-2-yl]butyl}(1,3-di(3-sulfopropyl)-1H-benzo[d]imidazol-3-ium) hydrogen sulfate* **8**

SAIL **8** was prepared from SAIL **7** (2.4 g, 4.2 mmol) and sulfuric acid (0.24 mL, 4.3 mmol) acording to the procedure for the preparation of SAIL **2**, to yield 2.8 g (100%) of solid product; **mp** 222–231 °C; **^1^H-NMR** (D_2_O, 400 MHz) 1.90 (2H, q, benzimidazole-CH_2_-CH_2_-C*H*_2_-CH_2_-benzimidazole) 2.06 (2H, q, benzimidazole-CH_2_-C*H*_2_-C*H*_2_-CH_2_-benzimidazole), 2.20 (6H, q, NCH_2_C*H*_2_CH_2_), 2.92 (6H, m, NCH_2_CH_2_C*H*_2_), 3.24 (2H, t, *J* = 8.0 Hz, -C*H*_2_-benzimidazole), 3.30 (2H, t, *J* = 8.0 Hz, -C*H*_2_-benzimidazole), 4.54 (6H, m, NC*H*_2_CH_2_CH_2_), 7.46 (2H, m, C*H*=C*H*, C5′, C6′ in benzimidazole), 7.52 (2H, m, C*H*=C*H*, C5, C6 in benzimidazole), 7.60 (1H, m, -C*H*=, C4′ in benzimidazole), 7.70 (1H, m, -C*H*=, C7′ in benzimidazole), 7.74 (2H, m, -C*H*=, C4, C7 in benzimidazole); **^13^C-NMR** (D_2_O, 100 MHz) 23.11 (s, benzimidazole-CH_2_-*C*H_2_-CH_2_-CH_2_-benzimidazole), 24.14 (s, benzimidazole-CH_2_-CH_2_-*C*H_2_-CH_2_-benzimidazole), 24.36 (s, NCH_2_*C*H_2_CH_2_), 24.55 (s, NCH_2_*C*H_2_CH_2_), 25.88 (s, benzimidazole-*C*H_2_-CH_2_-CH_2_-CH_2_-benzimidazole), 26.51 (s, benzimidazole-CH_2_-CH_2_-CH_2_-*C*H_2_-benzimidazole), 43.32 (s, N’CH_2_CH_2_*C*H_2_), 43.87 (s, NCH_2_CH_2_*C*H_2_), 47.46 (s, N’*C*H_2_CH_2_CH_2_), 47.54 (s, N*C*H_2_CH_2_CH_2_), 112.41 (s, -*C*H=, C7′ in benzimidazole), 112.77 (s, -*C*H=, C4, C7 in benzimidazole), 113.84 (s, -*C*H=, C4′in benzimidazole), 126.14 (s, -*C*H=, C6′ in benzimidazole), 126.47 (s, *C*H=*C*H, C5, C6 in benzimidazole), 126.77 (s, -*C*H=, C5′ in benzimidazole), 129.97 (s, >*C*H= in benzimidazole), 131.13 (s, >*C*H= in benzimidazole), 131.64 (s, >*C*H= in benzimidazole), 152.17 (s, HN-*C*=N, C2 in benzimidazole), 152.76 (s, HN-*C*=N, C2 in benzimidazole); ESI **MS** *m*/*z* [M+H]^+^ 657.20 (100), calcd for C_27_H_37_N_4_O_9_S_3_: 657.17; **IR** (ATR, cm^−1^) 3429 (C-H, benzimidazolium), 2940 (C-H, C_4_H_8_), 1635, 1559, 1518, 1469 (C=C, C=N, benzimidazolium), 1151, 1034 (S=O, -SO_3_H, HSO_4_^−^).

### 3.4. Esterification of Oleic Acid with Ethanol Catalyzed by Ionic Liquids and Analysis

#### 3.4.1. General Procedure for the Microwave Esterification Reaction

Esterification experiments were performed using a microwave reactor Plazmatronika (microwave power range 0 to 700 W) equipped with a condenser, temperature control system, and magnetic stirrer. During the experiment, a temperature sensor monitored the temperature in the reaction vessel and constant stirring was applied for homogeneous heating in all runs. 

Amounts of the fatty acids mixture (0.32 mL, 1 mmol), ethanol (1 mL, 16.5 mmol), and SAIL catalyst (varied from 0.1 to 0.3 mmol, the concentration range of SAIL was 0.1; 0.2; 0.3 mol/L) were charged into a 25 mL round bottom flask. After mixing, the first sample was taken and put on the TLC plate. Subsequently, the mixture was heated in the microwave reactor at 100% power (700 W) over 3 cycles, and the reaction temperature, measured by a sensor control system, was up to 100 °C, each cycle lasted 10 min and after each cycle, a sample was taken and applied on the TLC. The TLC plate was then eluted in a 9:1 hexane:diethyl ether system, and detected with a saturated solution of phosphomolybdic acid in a mixture of water: ethanol 1:1. After completion of the reaction in a microwave reactor, the conversions were determined by HPLC MS.

#### 3.4.2. Esterification of Fatty Acids with Ethanol with Conventional Heating 

A fixed amount of fatty acids (0.32 mL, 1 mmol), ethanol (1 mL, 16.5 mmol), and SAIL catalyst (0.3 mmol) was added to a 25 mL round bottom flask. After mixing, a first sample was taken and applied on the TLC plate, and the reaction was performed with magnetic stirring, heating in an oil bath at 100 °C under reflux condensation for 8 h. Every 2 h, a sample was taken and applied on the TLC plate. The plate was then eluted in a 9:1 n-hexane:diethylether system, and detected with a saturated solution of phosphomolybdic acid in a mixture of water:ethanol 1:1. After completion of the reaction, the conversions were determined by HPLC MS.

#### 3.4.3. The Recycling Experiments

The esterification reactions were carried out under the same conditions as in Section 3.4.2. A mixture of fatty acids (0.32 mL, 1 mmol), ethanol (1 mL, 16.5 mmol), and SAIL catalyst (0.3 mmol) was heated in an oil bath at 100 °C under reflux condensation for 2 h. Then the reaction mixture was cooled down in an ice bath and decanted. The upper layer was removed and stored for further analysis. Ethanol (1 mL) and fatty acids (0.32 mL) were added to the residue and the same procedure was repeated 4 times more.

## 4. Conclusions

In summary, the characterization results of FT-IR and NMR revealed that the three sulfonic acid group functionalized ionic liquids (SAIL) with a different number of catalytic groups and lipophilicity have been successfully synthesized. The highest catalytic performance for the conversion of fatty acids into respective fatty acids ethyl esters (FAEE) under microwave heating showed **SAIL 5**, containing a long lipophilic C15 chain. The catalytic activity was slightly higher than using **SAIL 2**, whereas **SAIL 8** was surprisingly almost inactive. With conventional heating (molar ratio ethanol to fatty acids 16.5:1, catalyst amount 30% based on the acid oleic molar amount, reaction time 2 h, and reaction temperature of the bath 100 °C), the conversions to biodiesel were about 98%. The conversion of both fatty acids to FAEE gradually decreases in repeated trials only in the case of **SAILs 2** and **5**. The catalytic activity of **SAIL 8** remains stable, the IL is the most effective catalyst for the esterification of fatty acids to FAEE at conventional heating. The SAIL catalysts can still maintain high catalytic activity after five cycles.

## Data Availability

The datasets generated during and/or analyzed during the current study are available from the corresponding author on reasonable request.

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
