# Peer review of "Design and Synthesis of New Sulfonic Acid Functionalized Ionic Liquids as Catalysts for Esterification of Fatty Acids with Bioethanol"

_molecules, 2023, doi:10.3390/molecules28135231_

Round 1
Reviewer 1 Report
In this paper, three types of sulfonic acid group functionalized ionic liquids (SAILs) with different number of catalytic groups and lipophilicity were synthesized and used as catalysts for the esterification of oleic acid with ethanol. The paper does not highlight the innovation of the research. Many researchers have done the similar work according to synthesizing of biodiesel catalyzed by ionic liquids, resulting in good results. And there are no compared results with those previously reported. Moreover, the presentations and quality of the manuscript should be improved. It is not refined, and it is wordy and repetitive. I am sorry that I cannot recommend this paper to be accepted by the journal of Molecules.
1. The author should provide necessary reaction conditions in testing the catalytic performance.
2. The author emphasizes the influence of the lipophilic of ionic liquid on its catalytic performance. Please add the lipophilic of different ionic liquids.
3. The necessary characterization and structural characteristics of different ionic liquids should be provided and analyzed, such as acid strength, thermodynamic stability as well as viscosity.
It is noted that this manuscript needs careful editing by someone with expertise in technical English editing paying particular attention to English grammar, spelling and sentence structure so that the goals and results of the review are clear to the readers.
Reviewer 2 Report
Here the authors reported that several SAILs with different number sulfonic acid groups and alkyl group were synthesized and characterized. The experimental results indicated that lipophilic alkyl chain of SAIL performed best activity for esterification reaction. The microwave heating was found to be more effective than conventional heating. But several areas need improvements before this manuscript can be considered for publication. See the comments below:
1. The innovation of catalyst design is necessary to demonstrate, there are various catalysts in the literature exploring influence of lipophilic alkyl chain (See Appl. Catal. A, 2021, 625, 118335; Energy, 2021, 219, 119637; Chin. J. Catal., 2015, 36, 982; Catal. Sci. Technol., 2013, 4, 1102). As discussed in manuscript, ILs are important active catalysts for biodiesel production that the authors should include in their comparison (Renew. Energy, 2022, 181, 341; Green Chem., 2019, 21, 3182; Catal. Today, 2020, 346, 87; ACS Sustain. Chem. Eng., 2020, 8, 6956; Green Energy Environ., 2021, 6, 271; Fuel, 2022, 307, 121876; Chem. Eng. Res. Design, 2022, 180, 134; Front. Chem., 2022, 10, 999607). Overall, a more careful analysis of the literature is required in the manuscript, because the references are outdated.
2. The ratio of ethanol to oil was 16.5:1, the influence of ratio is suggested to be provided in the paper, because the previous literature reported that low ratio was benefit to this reaction, the concentration of oil and catalyst was diluted at high ration.
3. In Table 1, SAIL 2 exhibited 86.8% conversion, by contrast only 4.3% was detected with the similar catalyst, the results are confusing.
4. The reaction temperature was 100 °C, at this point, which is higher than the boiling point of ethanol, how to control? It was suggested that the effect of temperature could provided.
5. The microwave heating was more effective than conventional heating, the microwave power range 0 to 700 W, why the power of 700 W was selected?
6. In rancid oil, there is not only oleic acid, but also other fatty acids, such as palmitic acid, stearic acid etc. How about the optimizing catalyst for these fatty acids?
Minor editing of English language required.
Reviewer 3 Report
The manuscript by ÄŒermák and colleagues described the synthesis of New Sulfonic Acid Functionalized Ionic Liquids as Catalysts for the Esterification of Fatty Acids with Bioethanol. Nevertheless, It can be improved by addressing the following suggestions:
1. Please assign numbers to every molecule in schemes
2. Please integrate proton peaks in the NMR. All the NMR spectra need to be properly processed. Also, enlarge the NMR spectra, better to have only two spectra per page.
3. Also, please provide the software files for NMRs
4. Provide the mobile phase with composition and Rf used for flash chromatography in the experimental procedures.
5. Label the IR peaks
6. Please add the HRMS/MS spectra in the supplementary file.
7. Please redraw all the schemes as per the journal’s guidelines.
8. Using old literature, please compare Catalytic Activity over Different (SAILs) in a tabular form with proper referencing.
9. Please describe the mechanism behind this SAILs catalysis.
10. The result and discussion section lacks clarity, it should explain why this particular catalyst showing such good results.
The authors composed this manuscript well with good coherence and flow.
Round 2
Reviewer 1 Report
The authors have revised the manuscript according to the comments. However, further revision is necessary. The main comments are as follows.
1. The author emphasizes the influence of the lipophilicity of ionic liquid on its catalytic performance. Please add the oil-water partition coefficients of different ionic liquids.
2. The comparison on the catalytic performance of some reported ionic liquid catalysts has been added in Table 3. However, these comparisons do not make some sense. Please add the turnover frequency (TOF) values.
3. The necessary characterization and structural characteristics of different ionic liquids should be provided and analyzed, such as acid strength as well as viscosity.
Moderate editing of English language is required.
Author Response
- The author emphasizes the influence of the lipophilicity of ionic liquid on its catalytic performance. Please add the oil-water partition coefficients of different ionic liquids.
Although SAIL5, which contains a long alkyl chain, is indeed a better catalyst than SAIL2, the best results are obtained with SAIL8, so lipophilicity is not the only important factor. Measuring the oil-water partition coefficients takes time and, in our opinion, does not make much sense with only three compounds to compare. It would only be useful if we had a series of compounds with different alkyl chain lengths, as reported e.g. in [30].
- The comparison on the catalytic performance of some reported ionic liquid catalysts has been added in Table 3. However, these comparisons do not make some sense. Please add theturnover frequency (TOF) values.
TOF values are usually not reported in the literature when authors compare catalytic activity in the field of biofuels (see e.g. [6,9,14,25]). Nevertheless, these values have been included in Table 3 because they can be easily calculated from the data in Table 3. However, in our opinion, this is not necessary as it is not only the TOF in a single batch experiment that determines the performance of a particular catalyst. If the catalyst can be recycled and the process made continuous, the lifetime of the catalyst is crucial.
- The necessary characterization and structural characteristics of different ionic liquids should be provided and analyzed, such as acid strength as well as viscosity.
In our opinion, there exists no correlation between the viscosity of a pure ionic liquid and its catalytic performance under experimental conditions of catalysis. The compounds used as catalysts are otherwise clearly characterised in our manuscript.
On the other hand, there should be a correlation between the acidity of the catalysts and their activity. Therefore, we performed additional experiments - acidity measurements using the Hammett function according to the literature [16]. However, the ionic liquids are not soluble in either water or pure ethanol to the required concentrations. Measurements in a methanol/water mixture (1/1, v/v) gave inexplicable results: the molar absorptivity of the protonated form was the same or even higher than that of pure p-nitroaniline (see the newly added document: Acidity_Hammett.docx). The same behaviour was observed for p-toluenesulfonic acid. Therefore, the determination of the acid strength was not possible in our hands.
Reviewer 2 Report
The authors have addressed most of my questions basically in a satisfactory manner. The referee is in support of its publication.
Author Response
We thank the reviewer for comments and recommendations which improved the quality of our manuscript.
Reviewer 3 Report
The authors have incorporated most of the suggestions and corrected the Nmr files and schemes. In my opinion, the manuscript can be accepted.
Author Response

(The authors gave the same response as above.)
